# GibbsNet: Iterative Adversarial Inference for Deep Graphical Models

Alex Lamb          R Devon Hjelm          Yaroslav Ganin          Joseph Paul Cohen

Aaron Courville                    Yoshua Bengio

## Abstract

Directed latent variable models that formulate the joint distribution as $p(x, z) = p(z)p(x \mid z)$ have the advantage of fast and exact sampling. However, these models have the weakness of needing to specify $p(z)$, often with a simple fixed prior that limits the expressiveness of the model. Undirected latent variable models discard the requirement that $p(z)$ be specified with a prior, yet sampling from them generally requires an iterative procedure such as blocked Gibbs-sampling that may require many steps to draw samples from the joint distribution $p(x, z)$. We propose a novel approach to learning the joint distribution between the data and a latent code which uses an adversarially learned iterative procedure to gradually refine the joint distribution, $p(x, z)$, to better match with the data distribution on each step. GibbsNet is the best of both worlds both in theory and in practice. Achieving the speed and simplicity of a directed latent variable model, it is guaranteed (assuming the adversarial game reaches the virtual training criteria global minimum) to produce samples from $p(x, z)$ with only a few sampling iterations. Achieving the expressiveness and flexibility of an undirected latent variable model, GibbsNet does away with the need for an explicit $p(z)$ and has the ability to do attribute prediction, class-conditional generation, and joint image-attribute modeling in a single model which is not trained for any of these specific tasks. We show empirically that GibbsNet is able to learn a more complex $p(z)$ and show that this leads to improved inpainting and iterative refinement of $p(x, z)$ for dozens of steps and stable generation without collapse for thousands of steps, despite being trained on only a few steps.

## 1  Introduction

Generative models are powerful tools for learning an underlying representation of complex data. While early undirected models, such as Deep Boltzmann Machines or DBMs (Salakhutdinov and Hinton, 2009), showed great promise, practically they did not scale well to complicated high-dimensional settings (beyond MNIST), possibly because of optimization and mixing difficulties (Bengio et al., 2012). More recent work on Helmholtz machines (Bornschein et al., 2015) and on variational autoencoders (Kingma and Welling, 2013) borrow from deep learning tools and can achieve impressive results, having now been adopted in a large array of domains (Larsen et al., 2015).

Many of the important generative models available to us rely on a formulation of some sort of stochastic latent or hidden variables along with a generative relationship to the observed data. Arguably the simplest is the *directed graphical models* (such as the VAE) with a factorized decomposition $p(z, x) = p(z)p(x \mid z)$. In this, it is typical to assume that $p(z)$ follows some factorized prior with simple statistics (such as Gaussian). While sampling with directed models is simple, inference and learning tends to be difficult and often requires advanced techniques such as approximate inference using a proposal distribution for the true posterior.

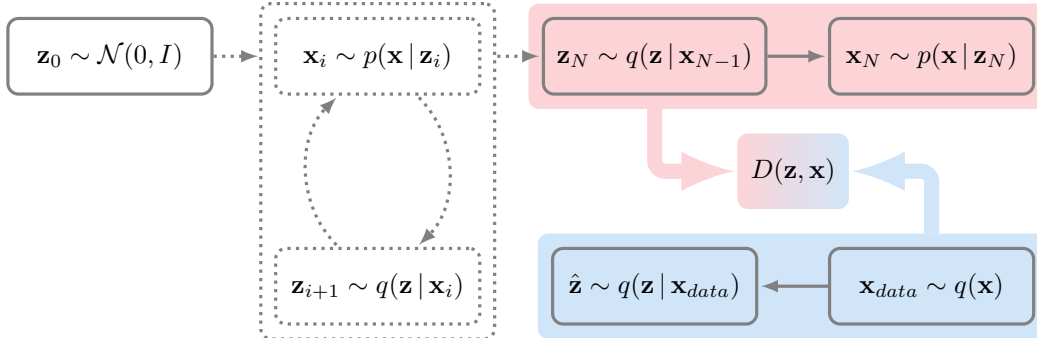

Figure 1: Diagram illustrating the training procedure for GibbsNet. The **unclamped chain** (dashed box) starts with a sample from an isotropic Gaussian distribution $\mathcal{N}(0, I)$ and runs for $N$ steps. The last step (iteration $N$) shown as a solid pink box is then compared with a single step from the **clamped chain** (solid blue box) using joint discriminator $D$.

The other dominant family of graphical models are *undirected graphical models*, such that the joint is represented by a product of clique potentials and a normalizing factor. It is common to assume that the clique potentials are positive, so that the un-normalized density can be represented by an energy function, $E$ and the joint is represented by $p(x, z) = e^{-E(z,x)}/Z$, where $Z$ is the normalizing constant or partition function. These so-called energy-based models (of which the Boltzmann Machine is an example) are potentially very flexible and powerful, but are difficult to train in practice and do not seem to scale well. Note also how in such models, the marginal $p(z)$ can have a very rich form (as rich as that of $p(x)$).

The methods above rely on a fully parameterized joint distribution (and approximate posterior in the case of directed models), to train with approximate maximum likelihood estimation (MLE, Dempster et al., 1977). Recently, generative adversarial networks (GANs, Goodfellow et al., 2014) have provided a likelihood-free solution to generative modeling that provides an implicit distribution unconstrained by density assumptions on the data. In comparison to MLE-based latent variable methods, generated samples can be of very high quality (Radford et al., 2015), and do not suffer from well-known problems associated with parameterizing noise in the observation space (Goodfellow, 2016). Recently, there have been advances in incorporating latent variables in generative adversarial networks in a way reminiscent of Helmholtz machines (Dayan et al., 1995), such as adversarially learned inference (Dumoulin et al., 2017; Donahue et al., 2017) and implicit variational inference (Huszár, 2017).

These models, as being essentially complex directed graphical models, rely on approximate inference to train. While potentially powerful, there is good evidence that using an approximate posterior necessarily limits the generator in practice (Hjelm et al., 2016; Rezende and Mohamed, 2015). In contrast, it would perhaps be more appropriate to start with inference (encoder) and generative (decoder) processes and derive the prior directly from these processes. This approach, which we call GibbsNet, uses these two processes to define a transition operator of a Markov chain similar to Gibbs sampling, alternating between sampling observations and sampling latent variables. This is similar to the previously proposed generative stochastic networks (GSNs, Bengio et al., 2013) but with a GAN training framework rather than minimizing reconstruction error. By training a discriminator to place a decision boundary between the data-driven distribution (with $x$ clamped) and the free-running model (which alternates between sampling $x$ and $z$), we are able to train the model so that the two joint distributions $(x, z)$ match. This approach is similar to Gibbs sampling in undirected models, yet, like traditional GANs, it lacks the strong parametric constraints, i.e., there is no explicit energy function. While losing some the theoretical simplicity of undirected models, we gain great flexibility and ease of training. In summary, our method offers the following contributions:

- We introduce the theoretical foundation for a novel approach to learning and performing inference in deep graphical models. The resulting model of our algorithm is similar to undirected graphical models, but avoids the need for MLE-based training and also lacks an explicitly defined energy, instead being trained with a GAN-like discriminator.

- We present a stable way of performing inference in the adversarial framework, meaning that useful inference is performed under a wide range of architectures for the encoder and decoder networks. This stability comes from the fact that the encoder $q(z \mid x)$ appears in both the clamped and the unclamped chain, so gets its training signal from both the discriminator in the clamped chain and from the gradient in the unclamped chain.

- We show improvements in the quality of the latent space over models which use a simple prior for $p(z)$. This manifests itself in improved conditional generation. The expressiveness of the latent space is also demonstrated in cleaner inpainting, smoother mixing when running blocked Gibbs sampling, and better separation between classes in the inferred latent space.

- Our model has the flexibility of undirected graphical models, including the ability to do label prediction, class-conditional generation, and joint image-label generation in a single model which is not explicitly trained for any of these specific tasks. To our knowledge our model is the first model which combines this flexibility with the ability to produce high quality samples on natural images.

## 2 Proposed Approach: GibbsNet

The goal of GibbsNet is to train a graphical model with transition operators that are defined and learned directly by matching the joint distributions of the model expectation with that with the observations clamped to data. This is analogous to and inspired by undirected graphical models, except that the transition operators, which correspond to blocked Gibbs sampling, are defined to move along a defined energy manifold, so we will make this connection throughout our formulation.

We first explain GibbsNet in the simplest case where the graphical model consists of a single layer of observed units and a single layer of latent variable with stochastic mappings from one to the other as parameterized by arbitrary neural network. Like Professor Forcing (Lamb et al., 2016), GibbsNet uses a GAN-like discriminator to make two distributions match, one corresponding to the model iteratively sampling both observation, $x$, and latent variables, $z$ (free-running), and one corresponding to the same generative model but with the observations, $x$, clamped. The free-running generator is analogous to Gibbs sampling in Restricted Boltzmann Machines (RBM, Hinton et al., 2006) or Deep Boltzmann Machines (DBM, Salakhutdinov and Hinton, 2009). In the simplest case, the free-running generator is defined by conditional distributions $q(z|x)$ and $p(x|z)$ which stochastically map back and forth between data space $x$ and latent space $z$.

To begin our free-running process, we start the chain with a latent variable sampled from a normal distribution: $z \sim \mathcal{N}(0, I)$, and follow this by $N$ steps of alternating between sampling from $p(x|z)$ and $q(z|x)$. For the clamped version, we do simple ancestral sampling from $q(z|x)$, given $x_{data}$ is drawn from the data distribution $q(x)$ (a training example). When the model has more layers (e.g., a hierarchy of layers with stochastic latent variables, à la DBM), the data-driven model also needs to iterate to correctly sample from the joint. While this situation highly resembles that of undirected graphical models, GibbsNet is trained adversarially so that its free-running generative states become indistinguishable from its data-driven states. In addition, while in principle undirected graphical models need to either start their chains from data or sample a very large number of steps, we find in practice GibbsNet only requires a very small number of steps (on the order of 3 to 5 with very complex datasets) from noise.

An example of the free-running (unclamped) chain can be seen in Figure 2. An interesting aspect of GibbsNet is that we found that it was enough and in fact best experimentally to back-propagate discriminator gradients *through a single step of the iterative procedure*, yielding more stable training. An intuition for why this helps is that each step of the procedure is supposed to generate increasingly realistic samples. However, if we passed gradients through the iterative procedure, then this gradient could encourage the earlier steps to store features which have downstream value instead of immediate realistic $x$-values.

### 2.1 Theoretical Analysis

We consider a simple case of an undirected graph with single layers of visible and latent units trained with alternating 2-step ($p$ then $q$) unclamped chains and the asymptotic scenario where the GAN objective is properly optimized. We then ask the following questions: in spite of training for a

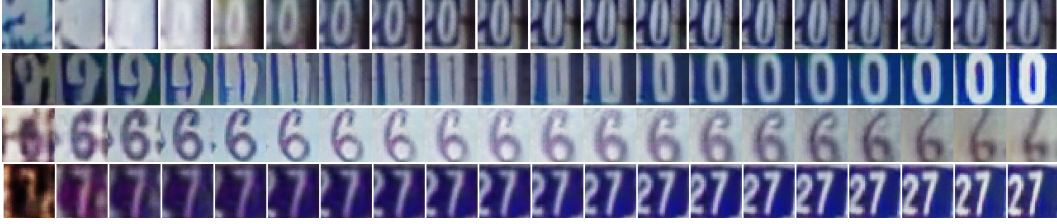

Figure 2: Evolution of samples for 20 iterations from the unclamped chain, trained on the SVHN dataset starting on the left and ending on the right.

bounded number of Markov chain steps, are we learning a transition operator? Are the encoder and decoder estimating compatible conditionals associated with the stationary distribution of that transition operator? We find positive answers to both questions.

A high level explanation of our argument is that if the discriminator is fooled, then the consecutive $(z, x)$ pairs from the chain match the data-driven $(z, x)$ pair. Because the two marginals on $x$ from these two distributions match, we can show that the next $z$ in the chain will form again the same joint distribution. Similarly, we can show that the next $x$ in the chain also forms the same joint with the previous $z$. Because the state only depends on the previous value of the chain (as it's Markov), then all following steps of the chain will also match the clamped distribution. This explains the result, validated experimentally, that even though we train for just a few steps, we can generate high quality samples for thousands or more steps.

**Proposition 1.** *If (a) the stochastic encoder $q(z|x)$ and stochastic decoder $p(x|z)$ inject noise such that the transition operator defined by their composition (p followed by q or vice-versa) allows for all possible x-to-x or z-to-z transitions ($x \rightarrow z \rightarrow x$ or $z \rightarrow x \rightarrow z$), and if (b) those GAN objectives are properly trained in the sense that the discriminator is fooled in spite of having sufficient capacity and training time, then (1) the Markov chain which alternates the stochastic encoder followed by the stochastic decoder as its transition operator $T$ (or vice-versa) has the data-driven distribution $\pi_D$ as its stationary distribution $\pi_T$, (2) the two conditionals $q(z|x)$ and $p(x|z)$ converge to compatible conditionals associated with the joint $\pi_D = \pi_T$.*

*Proof.* When the stochastic decoder and encoder inject noise so that their composition forms a transition operator $T$ with paths with non-zero probability from any state to any other state, then $T$ is ergodic. So condition (a) implies that $T$ has a stationary distribution $\pi_T$. The properly trained GAN discriminators for each of these two steps (condition (b)) forces the matching of the distributions of the pairs $(z_t, x_t)$ (from the generative trajectory) and $(x, z)$ with $x \sim q(x)$, the data distribution and $z \sim q(z \mid x)$, both pairs converging to the same data-driven distribution $\pi_D$. Because $(z_t, x_t)$ has the same joint distribution as $(z, x)$, it means that $x_t$ has the same distribution as $x$. Since $z \sim q(z \mid x)$, when we apply $q$ to $x_t$, we get $z_{t+1}$ which must form a joint $(z_{t+1}, x_t)$ which has the same distribution as $(z, x)$. Similarly, since we just showed that $z_{t+1}$ has the same distribution as $z$ and thus the same as $z_t$, if we apply $p$ to $z_{t+1}$, we get $x_{t+1}$ and the joint $(z_{t+1}, x_{t+1})$ must have the same distribution as $(z, x)$. Because the two pairs $(z_t, x_t)$ and $(z_{t+1}, x_{t+1})$ have the same joint distribution $\pi_D$, it means that the transition operator $T$, that maps samples $(z_t, x_t)$ to samples $(z_{t+1}, x_{t+1})$, maps $\pi_D$ to itself, i.e., $\pi_D = \pi_T$ is both the data distribution and the stationary distribution of $T$ and result (1) is obtained. Now consider the "odd" pairs $(z_{t+1}, x_t)$ and $(z_{t+2}, x_{t+1})$ in the generated sequences. Because of (1), $x_t$ and $x_{t+1}$ have the same marginal distribution $\pi_D(x)$. Thus when we apply the same $q(z|x)$ to these $x$'s we obtain that $(z_{t+1}, x_t)$ and $(z_{t+2}, x_{t+1})$ also have the same distribution. Following the same reasoning as for proving (1), we conclude that the associated transition operator $T_{\text{odd}}$ has also $\pi_D$ as stationary distribution. So starting from $z \sim \pi_D(z)$ and applying $p(x \mid z)$ gives an $x$ so that the pair $(z, x)$ has $\pi_D$ as joint distribution, i.e., $\pi_D(z, x) = \pi_D(z)p(x \mid z)$. This means that $p(x \mid z) = \frac{\pi_D(x,z)}{\pi_D(z)}$ is the $x \mid z$ conditional of $\pi_D$. Since $(z_t, x_t)$ also converges to joint distribution $\pi_D$, we can apply the same argument when starting from an $x \sim \pi_D(x)$ followed by $q$ and obtain that $\pi_D(z, x) = \pi_D(x)q(z \mid x)$ and so $q(z|x) = \frac{\pi_D(z,x)}{\pi_D(x)}$ is the $z \mid x$ conditional of $\pi_D$. This proves result (2). $\qquad\square$

## 2.2 Architecture

GibbsNet always involves three networks: the inference network $q(z|x)$, the generation network $p(x|z)$, and the joint discriminator. In general, our architecture for these networks closely follow Dumoulin et al. (2017), except that we use the boundary-seeking GAN (BGAN, Hjelm et al., 2017) as it explicitly optimizes on matching the opposing distributions (in this case, the model expectation and the data-driven joint distributions), allows us to use discrete variables where we consider learning graphs with labels or discrete attributes, and worked well across our experiments.

## 3 Related Work

**Energy Models and Deep Boltzmann Machines**   The training and sampling procedure for generating from GibbsNet is very similar to that of a deep Boltzmann machine (DBM, Salakhutdinov and Hinton, 2009): both involve blocked Gibbs sampling between observation- and latent-variable layers. A major difference is that in a deep Boltzmann machine, the "decoder" $p(x|z)$ and "encoder" $p(z|x)$ exactly correspond to conditionals of a joint distribution $p(x, z)$, which is parameterized by an energy function. This, in turn, puts strong constraints on the forms of the encoder and decoder.

In a restricted Boltzmann machine (RBM, Hinton, 2010), the visible units are conditionally independent given the hidden units on the adjacent layer, and likewise the hidden units are conditionally independent given the visible units. This may force the layers close to the data to need to be nearly deterministic, which could cause poor mixing and thus make learning difficult. These conditional independence assumptions in RBMs and DBMs have been discussed before in the literature as a potential weakness in these models (Bengio et al., 2012).

In our model, $p(x|z)$ and $q(z|x)$ are modeled by separate deep neural networks with no shared parameters. The disadvantage is that the networks are over-parameterized, but this has the added flexibility that these conditionals can be much deeper, can take advantage of all the recent advances in deep architectures, and have fewer conditional independence assumptions than DBMs and RBMs.

**Generative Stochastic Networks**   Like GibbsNet, generative stochastic networks (GSNs, Bengio et al., 2013) also directly parameterizes a transition operator of a Markov chain using deep neural networks. However, GSNs and GibbsNet have completely different training procedures. In GSNs, the training procedure is based on an objective that is similar to de-noising autoencoders (Vincent et al., 2008).

GSNs begin by drawing a sampling from the data, iteratively corrupting it, then learning a transition operator which de-noises it (i.e., reverses that corruption), so that the reconstruction after $k$ steps is brought closer to the original un-corrupted input.

In GibbsNet, there is no corruption in the visible space, and the learning procedure never involves "walk-back" (de-noising) towards a real data-point. Instead, the processes from and to data are modeled by different networks, with the constraint of the marginal, $p(x)$, matches the real distribution imposed through the GAN loss on the joint distributions from the clamped and unclamped phases.

**Non-Equilibrium Thermodynamics**   The Non-Equilibrium Thermodynamics method (Sohl-Dickstein et al., 2015) learns a reverse diffusion process against a forward diffusion process which starts from real data points and gradually injects noise until the data distribution matches a analytically tractible / simple distribution. This is similar to GibbsNet in that generation involves a stochastic process which is initialized from noise, but differs in that Non-Equilibrium Thermodynamics is trained using MLE and relies on noising + reversal for training, similar to GSNs above.

**Generative Adversarial Learning of Markov Chains**   The Adversarial Markov Chain algorithm (AMC, Song et al., 2017) learns a markov chain over the data distribution in the visible space. GibbsNet and AMC are related in that they both involve adversarial training and an iterative procedure for generation. However there are major differences. GibbsNet learns deep graphical models with latent variables, whereas the AMC method learns a transition operator directly in the visible space. The AMC approach involves running chains which start from real data points and repeatedly apply the transition operator, which is different from the clamped chain used in GibbsNet. The experiments

shown in Figure 3 demonstrate that giving the latent variables to the discriminator in our method has a significant impact on inference.

**Adversarially Learned Inference (ALI)**   Adversarially learned inference (ALI, Dumoulin et al., 2017) learns to match distributions generative and inference distributions, $p(x, z)$ and $q(x, z)$ (can be thought of forward and backward models) with a discriminator, so that $p(z)p(x \mid z) = q(x)q(z \mid x)$. In the single latent layer case, GibbsNet also has forward and reverse models, $p(x \mid z)$ and $q(z \mid x)$. The un-clamped chain is sampled as $p(z), p(x \mid z), q(z \mid x), p(x \mid z), \ldots$ and the clamped chain is sampled as $q(x), q(z \mid x)$. We then adversarially encourage the clamped chain to match the equilibrium distribution of the unclamped chain. When the number of iterations is set to $N = 1$, GibbsNet reduces to ALI. However, in the general setting of $N > 1$, Gibbsnet should learn a richer representation than ALI, as the prior, $p(z)$, is no longer forced to be the simple one at the beginning of the unclamped phase.

# 4   Experiments and Results

The goal of our experiments is to explore and give insight into the joint distribution $p(x, z)$ learned by GibbsNet and to understand how this joint distribution evolves over the course of the iterative inference procedure. Since ALI is identical to GibbsNet when the number of iterative inference steps is $N = 1$, results obtained with ALI serve as an informative baseline.

From our experiments, the clearest result (covered in detail below) is that the $p(z)$ obtained with GibbsNet can be more complex than in ALI (or other directed graphical models). This is demonstrated directly in experiments with 2-D latent spaces and indirectly by improvements in classification when directly using the variables $q(z \mid x)$. We achieve strong improvements over ALI using GibbsNet even when $q(z \mid x)$ has exactly the same architecture in both models.

We also show that GibbsNet allows for gradual refinement of the joint, $(x, z)$, in the sampling chain $q(z \mid x), p(x \mid z)$. This is a result of the sampling chain making small steps towards the equilibrium distribution. This allows GibbsNet to gradually improve sampling quality when running for many iterations. Additionally it allows for inpainting and conditional generation where the conditioning information is not fixed during training, and indeed where the model is not trained specifically for these tasks.

## 4.1   Expressiveness of GibbsNet's Learned Latent Variables

**Latent structure of GibbsNet**   The latent variables from $q(z \mid x)$ learned from GibbsNet are more expressive than those learned with ALI. We show this in two ways. First, we train a model on the MNIST digits 0, 1, and 9 with a 2-D latent space which allows us to easily visualize inference. As seen in Figure 3, we show that GibbsNet is able to learn a latent space which is not Gaussian and has a structure that makes the different classes well separated.

**Semi-supervised learning**   Following from this, we show that the latent variables learned by GibbsNet are better for classification. The goal here is not to show state of the art results on classification, but instead to show that the requirement that $p(z)$ be something simple (like a Gaussian, as in ALI) is undesirable as it forces the latent space to be filled. This means that different classes need to be packed closely together in that latent space, which makes it hard for such a latent space to maintain the class during inference and reconstruction.

We evaluate this property on two datasets: Street View House Number (SVHN, Netzer et al., 2011) and permutation invariant MNIST. In both cases we use the latent features $q(z \mid x)$ directly from a trained model, and train a 2-layer MLP on top of the latent variables, without passing gradient from the classifier through to $q(z \mid x)$. ALI and GibbsNet were trained for the same amount of time and with exactly the same architecture for the discriminator, the generative network, $p(x \mid z)$, and the inference network, $q(z \mid x)$.

On permutation invariant MNIST, ALI achieves 91% test accuracy and GibbsNet achieves 97.7% test accuracy. On SVHN, ALI achieves 66.7% test accuracy and GibbsNet achieves 79.6% test accuracy. This does not demonstrate a competitive classifier in either case, but rather demonstrates that the latent space inferred by GibbsNet keeps more information about its input image than the encoder

learned by ALI. This is consistent with the reported ALI reconstructions (Dumoulin et al., 2017) on SVHN where the reconstructed image and the input image show the same digit roughly half of the time.

We found that ALI's inferred latent variables not being effective for classification is a fairly robust result that holds across a variety of architectures for the inference network. For example, with 1024 units, we varied the number of fully-connected layers in ALI's inference network between 2 and 8 and found that the classification accuracies on the MNIST validation set ranged from 89.4% to 91.0%. Using 6 layers with 2048 units on each layer and a 256 dimensional latent prior achieved 91.2% accuracy. This suggests that the weak performance of the latent variables for classification is due to ALI's prior, and is probably not due to a lack of capacity in the inference network.

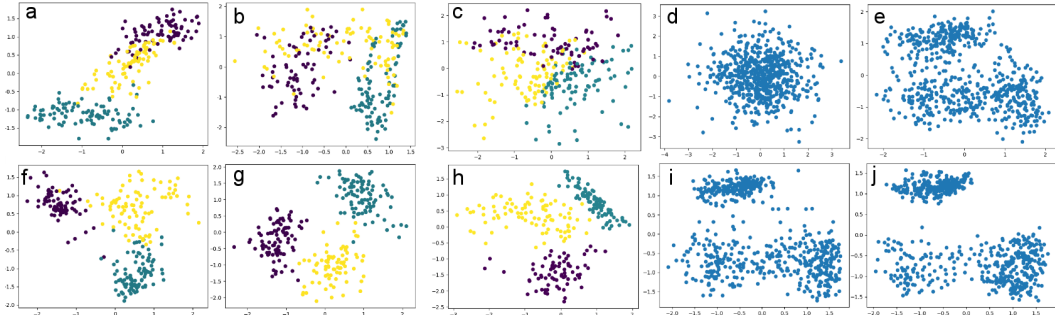

Figure 3: Illustration of the distribution over inferred latent variables for real data points from the MNIST digits (0, 1, 9) learned with different models trained for roughly the same amount of time: GibbsNet with a determinstic decoder and the latent variables not given to the discriminator (a), GibbsNet with a stochastic decoder and the latent variables not given to the discriminator (b), ALI (c), GibbsNet with a deterministic decoder (f), GibbsNet with a stochastic decoder with two different runs (g and h), GibbsNet with a stochastic decoder's inferred latent states in an unclamped chain at 1, 2 , 3, and 15 steps (d, e, i, and j, respectively) into the P-chain (d, e, i, and j, respectively). Note that we continue to see refinement in the marginal distribution of z when running for far more steps (15 steps) than we used during training (3 steps).

## 4.2 Inception Scores

The GAN literature is limited in terms of quantitative evaluation, with none of the existing techniques (such as inception scores) being satisfactory (Theis et al., 2015). Nonetheless, we computed inception scores on CIFAR-10 using the standard method and code released from Salimans et al. (2016). In our experiments, we compared the inception scores from samples from Gibbsnet and ALI on two tasks, generation and inpainting.

Our conclusion from the inception scores (Table 1) is that GibbsNet slightly improves sample quality but greatly improves the expressiveness of the latent space z, which leads to more detail being preserved in the inpainting chain and a much larger improvement in inception scores in this setting. The supplementary materials includes examples of sampling and inpainting chains for both ALI and GibbsNet which shows differences between sampling and inpainting quality that are consistent with the inception scores.

Table 1: Inception Scores from different models. Inpainting results were achieved by fixing the left half of the image while running the chain for four steps. Sampling refers to unconditional sampling.

| Source | Samples | Inpainting |
|---|---|---|
| Real Images | 11.24 | 11.24 |
| ALI (ours) | 5.41 | 5.59 |
| ALI (Dumoulin) | 5.34 | N/A |
| GibbsNet | 5.69 | 6.15 |

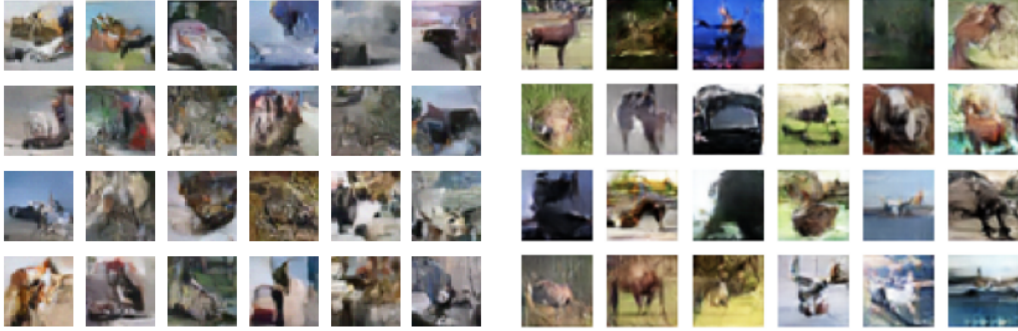

Figure 4: CIFAR samples on methods which learn transition operators. Non-Equilibrium Thermodynamics (Sohl-Dickstein et al., 2015) after 1000 steps (left) and GibbsNet after 20 steps (right).

### 4.3 Generation, Inpainting, and Learning the Image-Attribute Joint Distribution

**Generation**  Here, we compare generation on the CIFAR dataset against Non-Equilibrium Thermodynamics method (Sohl-Dickstein et al., 2015), which also begins its sampling procedure from noise. We show in Figure 4 that, even with a relatively small number of steps (20) in its sampling procedure, GibbsNet outperforms the Non-Equilibrium Thermodynamics approach in sample quality, even after many more steps (1000).

**Inpainting**  The inpainting that can be done with the transition operator in GibbsNet is stronger than what can be done with an explicit conditional generative model, such as Conditional GANs, which are only suited to inpainting when the conditioning information is known about during training or there is a strong prior over what types of conditioning will be performed at test time. We show here that GibbsNet performs more consistent and higher quality inpainting than ALI, even when the two networks share exactly the same architecture for $p(x \mid z)$ and $q(z \mid x)$ (Figure 5), which is consistent with our results on latent structure above.

**Joint generation**  Finally, we show that GibbsNet is able to learn the joint distribution between face images and their attributes (CelebA, Liu et al., 2015) (Figure 6). In this case, $q(z \mid x, y)$ ($y$ is the attribute) is a network that takes both the image and attribute, separately processing the two modalities before joining them into one network. $p(x, y \mid z)$ is one network that splits into two networks to predict the modalities separately. Training was done with continuous boundary-seeking GAN (BGAN, Hjelm et al., 2017) on the image side (same as our other experiments) and discrete BGAN on the attribute side, which is an importance-sampling-based technique for training GANs with discrete data.

## 5   Conclusion

We have introduced GibbsNet, a powerful new model for performing iterative inference and generation in deep graphical models. Although models like the RBM and the GSN have become less investigated in recent years, their theoretical properties worth pursuing, and we follow the theoretical motivations here using a GAN-like objective. With a training and sampling procedure that is closely related to undirected graphical models, GibbsNet is able to learn a joint distribution which converges in a very small number of steps of its Markov chain, and with no requirement that the marginal $p(z)$ match a simple prior. We prove that at convergence of training, in spite of unrolling only a few steps of the chain during training, we obtain a transition operator whose stationary distribution also matches the data and makes the conditionals $p(x \mid z)$ and $q(z \mid x)$ consistent with that unique joint stationary distribution. We show that this allows the prior, $p(z)$, to be shaped into a complicated distribution (not a simple one, e.g., a spherical Gaussian) where different classes have representations that are easily separable in the latent space. This leads to improved classification when the inferred latent variables $q(z|x)$ are used directly. Finally, we show that GibbsNet's flexible prior produces a flexible model which can simultaneously perform inpainting, conditional image generation, and prediction with a single model not explicitly trained for any of these specific tasks, outperforming a competitive ALI baseline with the same setup.

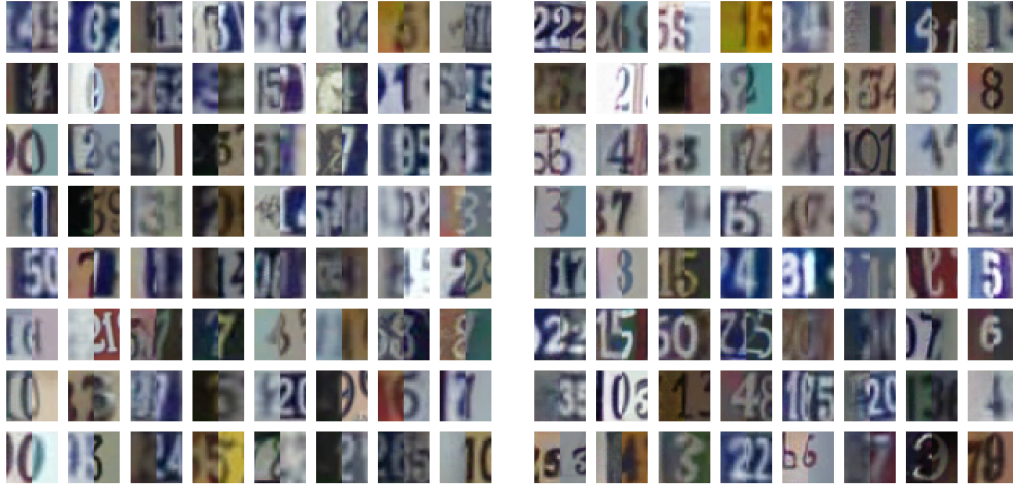

(a) SVHN inpainting after 20 steps (ALI).　　(b) SVHN inpainting after 20 steps (GibbsNet).

Figure 5: Inpainting results on SVHN, where the right side is given and the left side is inpainted. In both cases our model's trained procedure did not consider the inpainting or conditional generation task at all, and inpainting is done by repeatedly applying the transition operators and clamping the right side of the image to its observed value. GibbsNet's richer latent space allows the transition operator to keep more of the structure of the input image, allowing for tighter inpainting.

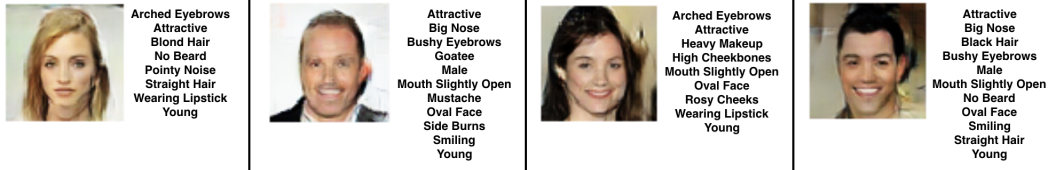

Figure 6: Demonstration of learning the joint distribution between images and a list of 40 binary attributes. Attributes (right) are generated from a multinomial distribution as part of the joint with the image (left).

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
