[Reviews · NeurIPS 2017]

Reviewer 1



This paper presents GibbsNet, a deep generative model formulated as transition operators. The transition operators are learned in an adversarial way, similar to that of the adversarially learned inference (ALI). However instead of using a fixed prior p(z), GibbsNet does not require the specification of a particular prior, but rather learn a prior implicitly. Training is done by unrolling the sampling process multiple times and doing adversarial learning to match the sampling distribution to the one clamped from data and doing posterior only once. When unrolling for only one step GibbsNet becomes equivalent to ALI. I found the idea of having an implicit prior quite interesting, and making it implicit does allow much more sophisticated priors to be used. Not requiring the prior to be Gaussian lifted some restrictions on the model and provided an opportunity to make the model more expressive. Compared to other transition operator-based models, GibbsNet has the appealing property that the transition process passes through a latent layer therefore may enable faster mixing, however this aspect is not explored in this paper. In terms of more technical details, I found the assumptions in Proposition 1 to be a little unrealistic, in particular assumption (a) will be hardly satisfied by any typical model we use. Usually, in latent variable models our latent variables z live in a much lower dimensionality space than visible variables x, therefore the mapping from z to x will only cover a small subspace of x. It is possible to make the probability of any x non-zero by using a noise distribution over x but purely relying on that is not going to be very effective. However on the other side the z-to-z transitions should be able to cover all the possibilities and hence learning complicated q(z) distributions is possible. This paper is mostly clear but has some writing issues, in particular almost none of the figures and therefore visualization results are directly referenced or discussed in the main text. Since the paper is not that tight on space, some more discussion about the experiment results should be included.

Reviewer 2



Summary- This paper proposes a novel and interesting way of training and doing inference in encoder-decoder style deep generative models. The transition operator, parametrized by encoder and decoder neural nets, starts from latent variables (z) that are noise, and produces a pair (x, z) by repeatedly applying itself. This generator model is trained to fool a discriminator that is trying to distinguish between generated samples, and (x, z) pairs computed on real data. In other words, instead of using the model's own Langevin dynamics to run a Markov Chain, a transition operator is trained to directly produce samples. This helps avoid the problems inherent in trying to sample from directed deep generative models (such as chains not mixing well), at the cost of not having a proper energy function, or probability distribution associated with the learned transition operator. Strengths- - The model is novel and interesting. It provides a way of training deep generative models that are amenable to feed-forward inference, feed-forward decoding, and can also generate samples in a few steps. - The paper is well-written and easy to follow. - The experiments are well-designed and help highlight the properties of the learned joint distribution. Relevant baselines are compared to. Weaknesses- - No major weaknesses. It would be good to have a quantitative evaluation of the generative performance, but that is understandably hard to do. Minor commenpt and typos- - "For the transition operator to produce smooth mixing, it is necessary for the latent variable q(z|x)." : Something missing here ? - In Fig 5, it might help compare results better if the same half-images were used for both ALI and GibbsNet. Overall The proposed model is novel and has many desirable properties. It provides a good example of using adversarial training to learn an interesting generative model. This idea seems quite powerful and can in future be applied on more challenging problems.

Reviewer 3



The authors proposed an extension over the Adverserially Learnt Inference(ALI) GAN that cycles between the latent and visible space for a few steps. The model iteratively refines the latent distribution by alternating the generator and approximate inference model in a chain computation. A joint distribution of both latent and visible data is then learnt by backpropagating through the iterative refinement process. The authors empirically demonstrated their model on a few imprint tasks. Strength: - The paper is well-organized and is easy to follow. - The experimental results on semi-supervised learning are encouraging. (more on that see the comment below. ) Weakness: - The main objection I have with the paper is that the authors did not put in any effort to quantitatively evaluate their newly proposed GAN training method. Comparing the inception score on CIFAR-10 with ALI and other benchmark GAN methods should be a must. The authors should also consider estimating the actual log likelihood of their GAN model by running the evaluation method proposed:"On the Quantitative Analysis of Decoder-Based Generative Models", Wu et al. The bottom line is that without appropriate quantitative analysis, it is hard to evaluate how well the proposed method does in general. What should help is to see a plot where an x-axis is the number of Gibbs steps and y-axis as one of the quantitative measures. - The improvement of the proposed method seems to be very marginal compared to ALI. The appropriate baseline comparison should be a deeper ALI model that has 2N number of layers. The "Gibbs chain" used throughout this paper is almost like a structured recurrent neural network with some of the intermediate hidden layers actually represents x and z. So, it is unfair to compare a 3-step GibbsNet with a 2-layer feedforward ALI model. =================== After I have read the rebuttal from the author, I have increased my score to reflect the new experiments conducted by the authors. The inception score results and architecture comparisons have addressed my previous concerns on evaluation. I am still concerned regarding the experimental protocols. The exact experimental setup for the semi-supervised learning results was not explained in detail. I suspect the GibbsNet uses a very different experimental protocol for SVHN and MNIST than the original ALI paper. It is hard to evaluate the relative improvement over ALI if the protocols are totally different. It is necessary to include all the experimental details in a future revision for reproducibility.